# Noise-Robust 3D Pose Estimation Using Appearance Similarity Based on the Distributed Multiple Views

**DOI:** 10.3390/s24175645

**Published:** 2024-08-30

**Authors:** Taemin Hwang, Minjoon Kim

**Affiliations:** 1Department of Autonomous Intelligent System Research Center, Korea Electronics Technology Institute, Seongnam 13509, Republic of Korea; taemin.hwang@keti.re.kr; 2Division of Semiconductor and Electronics Engineering, Hankuk University of Foreign Studies, Yongin 17035, Republic of Korea

**Keywords:** 3D pose estimation, multi-view geometry, on-device AI, bipartite graph matching

## Abstract

In this paper, we present a noise-robust approach for the 3D pose estimation of multiple people using appearance similarity. The common methods identify the cross-view correspondences between the detected keypoints and determine their association with a specific person by measuring the distances between the epipolar lines and the joint locations of the 2D keypoints across all the views. Although existing methods achieve remarkable accuracy, they are still sensitive to camera calibration, making them unsuitable for noisy environments where any of the cameras slightly change angle or position. To address these limitations and fix camera calibration error in real-time, we propose a framework for 3D pose estimation which uses appearance similarity. In the proposed framework, we detect the 2D keypoints and extract the appearance feature and transfer it to the central server. The central server uses geometrical affinity and appearance similarity to match the detected 2D human poses to each person. Then, it compares these two groups to identify calibration errors. If a camera with the wrong calibration is identified, the central server fixes the calibration error, ensuring accuracy in the 3D reconstruction of skeletons. In the experimental environment, we verified that the proposed algorithm is robust against false geometrical errors. It achieves around 11.5% and 8% improvement in the accuracy of 3D pose estimation on the Campus and Shelf datasets, respectively.

## 1. Introduction

Human pose estimation has a wide range of applications, including motion capture, surveillance, and augmented/virtual reality (AR/VR). Recent advancements in computer vision, driven by the rapid development of machine learning algorithms such as convolutional neural networks (CNN), have significantly improved human pose estimation [1]. This human pose estimation problem can be categorized into various subfields based on the factors such as number of individuals, resulting in the following subdivisions: single person 3D pose estimation [2,3], and multi-person 3D pose estimation [4,5,6]. In this paper, we address the problem of multi-person 3D pose estimation.

In addition, 3D pose estimation problems can be classified into either a single view [4,7,8] or multiple view system [5,6,9,10]. In multi-person 3D pose estimation problems, while approaches for estimating 3D poses using a single camera have been proposed, they continue to encounter challenges such as self-occlusion, motion blur, and so on. To mitigate these problems and enhance accuracy, 3D pose estimation across multiple views has been introduced. The multiple view system addresses the self-occlusion and motion blur by employing geometric information from other side views. Furthermore, multi-view systems also offer the distinct advantage of accurately representing 3D poses in a global coordinate system through global camera calibration.

In these multiple view systems, the greater the number of connected cameras, the more the complementary data become available. It leads to enhanced accuracy in the 3D pose estimation. However, this approach is constrained by the necessary bandwidth for transmitting substantial video data to the server. For instance, when using a USB camera, this system encounters limitations due to USB bandwidth constraints. For example, USB 3.1 Gen 1 offers a theoretical bandwidth of 5 Gbps [11]. This implies that connecting too many cameras to a single port using USB hubs can result in system failure due to bandwidth restriction [12]. The available bandwidth of 1000 base Ethernet is also limited to 1 Gbps [13].

In response to these limitations, researchers have explored edge processing techniques [14,15]. Edge AI devices, equipped with lightweight GPUs, can detect 2D human poses from RGB images. Instead of transmitting high-capacity RGB images, edge devices transmit low-capacity 2D human pose results to the server, effectively solving the bandwidth limitation issue. Furthermore, these systems also contribute to the structuring of a real-time system, reducing overall processing time as each device operates simultaneously. This paper also leverages the edge processing system as a fundamental component of our approach for the aforementioned benefits.

However, while edge processing can enhance accuracy and reduce execution time, the multiple view system remains sensitive to the accuracy of global camera calibration. Correct camera calibration is essential for achieving accurate and reliable results in instances that involve multiple view triangulation. If the camera calibration is incorrect or inaccurate, it can introduce projection errors and affect the overall quality of the triangulation results. In well-controlled laboratory environments, multiple cameras are tightly mounted on rigid frames. However, in most real-world scenarios, cameras may be mounted on simple tripods or could be installed outdoors. In such situations, the pose of a mounted camera is more vulnerable to external disturbances, including shock, vibration, gravity, or wind [16,17,18].

Thus, we propose a noise-robust 3D pose estimation method through camera calibration correction. The proposed system can adaptively adjust extrinsic parameters by detecting incorrect camera calibration through a comparison of geometrical affinity and appearance similarity. Intuitively, if it is judged to be the same person based on appearance of their clothing but, geometrically, it appears to be a different person, this could signal an issue with the calibration of a specific camera.

Initially, the edge device detects 2D human poses and extracts appearance features from the RGB image. Subsequently, both the 2D human pose data and appearance features are transmitted to the central server. The central server then matches 2D human poses across multiple views using both the geometrical affinity and appearance similarity. Subsequently, the central server compares the results. If the 2D human poses from a particular camera are excluded from the geometrical affinity group, the central server selects that camera for the updating of the extrinsic parameters. Given a 3D skeleton and a 2D skeleton from the miscalibrated camera, we can estimate the rotation matrix and translation vector. This process enables a noise-robust system, achieved by identifying cameras with calibration errors and correcting their calibration adaptively.

In the experimental evaluation, we simulate the environment where the calibration parameters of some of the cameras change considering the realistic conditions where camera calibration may drift or become inconsistent over time. In this scenario, we compare the accuracy of reconstructed 3D human poses across different methods. Through quantitative and qualitative results, including PCP (Percentage of Correct Parts) score comparisons and the visual inspection of restored 3D poses, we demonstrate that the proposed method improves the 3D pose estimation accuracy.

In summary, the primary objective of this paper is building a noise-robust system by using additional appearance features from conventional geometrical approaches. In the experimental results, we verified that only 48 bits of clothing color can enhance the 3D pose estimation accuracy. This approach could be useful in a practical distributed system consisting of edge devices, where minimal data are transmitted to the central server due to the limited bandwidth and computational resources.

The remainder of this paper is organized as follows: Section 2 describes the related works. In Section 3, we explain system architecture, while Section 4 introduces our noise-robust 3D pose estimation framework. Section 5 presents the experimental results and, finally, Section 6 concludes the paper.

## 2. Related Work

In recent years, 3D human pose estimation has seen significant advancements with improvements in deep learning techniques and computational resources. This section reviews the various approaches for 3D human pose estimation and edge AI approaches.

### 2.1. 3D Human Pose Estimation

In the rapid development of neural networks, which has led to remarkable progress in computer vision, various strategies have been proposed for estimating 3D human poses. DeepPose [19] first utilized a CNN model to solve the complexities of 2D human pose estimation and achieved a significant milestone with the introduction. After that, various researchers focused on deriving 3D poses [20,21,22]. For the single-person 3D human pose estimation, according to the regression steps, prior work can be divided into two main categories—the one-stage approach and the two-stage approach. One-stage approaches estimate the 3D poses directly from a given image. In contrast, two-stage approaches have two separate steps where a 2D pose is estimated first, and then the 2D pose is lifted to the 3D space. The two-stage approaches are known to have a “reducing the overfitting” effect, considering the quantity and the quality of the labeled 3D data [1].

As mentioned earlier, the one-stage approaches directly estimate the 3D pose without detecting the 2D poses. Li and Chan showed that deep neural networks estimate 3D human poses from an image with a reasonable accuracy [21]. They jointly trained pose regression tasks with multiple body part detection tasks, known as the heterogeneous multitask framework. However, the previous CNN-based techniques for estimating 3D human poses ignore the dependencies between the different body joint locations. To account for dependencies and results in increased performance, Tekin et al. [23] proposed a structured 3D human pose prediction framework from monocular images. Nevertheless, relying solely on a single viewpoint remains challenging because of occlusions, complex poses, or actions. To overcome these limitations and leverage complementary information from multiple perspectives, two-stage approaches were proposed.

These two-stage strategies solve the occlusion problem by employing a multi-view methodology. The multi-view methodologies use geometrical relationships between multi-views to estimate the 3D pose based on the 2D joint locations. The basic architecture of these models implements cascaded 2D and 3D estimation. Takahashi et al. [24] first proposed an algorithm of estimating the 3D human pose from multi-view videos captured by unsynchronized and non-calibrated cameras. In the proposed algorithm, a 2D pose detector estimates 2D poses from multi-view videos and then a median filter applies a cubic spline interpolation to the output data to reduce detection errors. Then, it selects two cameras which are initialized by the standard SfM (Structure from Motion) and decomposed into extrinsic parameters to estimate the 3D pose through triangulation. Also, A lightweight solution [25] is proposed using the Direct Linear Transform (DLT) layer. The DLT layer represents the association between the 3D and 2D poses from multi-views as a homogeneous linear system. Then, it resolves the 3D pose using singular value decomposition (SVD). Although remarkable solutions for single-person 3D pose estimation have been proposed, research on 3D multi-person pose estimation is still limited because matching 2D poses across multiple views is challenging. A typical approach is to use the epipolar constraint to verify if two 2D poses are projections of the same 3D pose for each pair of views [26]. Dong et al. [9] introduced a novel fast and robust method that is based on a multi-way matching algorithm that aims to cluster the detected 2D poses among multi-view images. Moreover, they combined geometric and appearance information for cross-view matching. Then, the 3D pose is separately predicted for each individual from the resulting matched 2D poses. In addition, Tanke et al. [27] introduced a simple approach to solve the association problem, using a bipartite matching algorithm to track multiple people over the multi-view images. Subsequent research has continued to explore 3D pose estimation using geometrical features. Remelli et al. [28] presented a lightweight solution for 3D pose recovery from multi-view images captured by spatially calibrated cameras. Their method leverages 3D geometry to fuse input images into a unified latent representation of pose, independent of camera viewpoints. Additionally, more recent work [29] has employed a cross-view U-shaped graph convolutional network (CV-UGCN) that performs triangulation to lift 2D keypoints into coarse 3D poses and refines them for improved accuracy.

### 2.2. Edge AI-Based 3D Human Pose Estimation

Traditionally, 3D human pose estimation requires high-performance computing resources; however, edge devices have recently become widely used for deploying AI models, with the recent advances in hardware performance and AI model optimization. ConvNeXtPose [30] introduced a mobile 3D human pose estimation model, achieving real-time performances. They achieved such state-of-the-art performances while requiring fewer parameters and FLOPS compared to the existing models. Hossain et al. [31] proposed RRMP, which performs 2D and 3D pose estimation in real-time for mobile edge computing devices. They verified that the RRMP performs with an average of 40 fps in real-time execution.

In addition, two-stage approaches with edge processing have also been proposed. Bultmann et al. [14] introduced a method for the estimation of 3D human poses from a multi-camera setup, employing distributed smart edge sensors. The smart edge sensors detect 2D poses and transfer it to the host to lift 2D poses to a 3D pose. They used the Google Edge TPU Dev Board (Google LLC, Mountain View, CA, USA) [32] equipped with an ARM Cortex-A53 quad-core processor as edge device and showed that the framework successfully works in the environment where 16 sensor boards were attached to the ceiling of their lab. There has also been an instance of commercial deployment. StereoLabs, San Francisco, USA has produced a vision-based AI product, releasing SDK which includes a Fusion API feature [15]. The Fusion API provides two different workflows—Local and Network. Local works in the traditional method. The host computer reads images from the multiple cameras, connected via a USB, and estimates the 3D human pose. On the other hand, in the network workflow, the system works with multiple clients and a single host computer. The client is an edge device such as NVIDIA Jetson Xavier which transfers the metadata, and then the host computer receives and aggregate it for 3D body tracking.

Obtaining and aggregating important features from the edge devices are important, but most traditional methods use only poses and multi-view camera parameters for matching persons. However, in this case, certain information which can be used to identify a person can be ignored such as clothing. In this paper, we propose an edge-based 3D pose estimation using appearance similarity combined with geometric knowledge.

## 3. System Architecture

The proposed framework consists of multiple edge AI devices and a single central server as shown in Figure 1. The edge devices are responsible for detecting 2D human poses and extracting appearance features from RGB images. Then, they transmit the 2D pose data and appearance features to the central server. In an ideal scenario without any calibration errors, the central server groups the 2D poses based on geometric affinity and reconstructs them into 3D poses using multi-view triangulation. However, if there are calibration errors present, the central server identifies any incorrect 2D poses by comparing the geometric affinity and appearance similarity of the 2D pose groups. With the reconstructed 3D poses and the identified incorrect 2D poses, the central server can estimate the rotation and translation vectors required to fix the calibration errors. These estimated vectors are feedback to the camera with the wrong calibration. Through this calibration self-correction method, the proposed framework achieves robustness in noisy environments with potential calibration errors by utilizing very little data corresponding to appearance similarity.

## 4. Proposed Framework

This section describes the distinct functions and roles of the edge devices and the central server within our proposed framework for noise-robust 3D pose estimation. The edge devices transmit the 2D human pose data and appearance features necessary for 3D pose estimation to the central server. The central server then groups the 2D human poses corresponding to each individual and detects any camera with an incorrect calibration to apply calibration error correction.

### 4.1. Edge Device

In the proposed framework, the edge devices have a role as the frontline processor. We consider that each edge device has a lightweight GPU, reads raw RGB images from camera, and extracts valuable information such as 2D human pose and appearance features. Then, the edge devices transmit these results to the central server. Through this approach, we can address the bandwidth requirements and lack of computational resources on the central system.

#### 4.1.1. 2D Human Pose Detection

First, the edge device detects a 2D human pose from the RGB image. Various 2D human pose detection models for edge AI have been proposed in recent years [14,31,33]. For our implementation, we utilized OpenPose (v1.7.0) [33] due to its well-established reputation and widespread adoption in the field. We denoted the detected 2D human pose as follows:(1)$${\left\{p_{i,\;k}\right\}}_{i\in [1,\;N]}^{k\in [1,\;K_{i}]}$$
where N is the number of edge devices, same for the calibrated cameras, and $K_{i}$ is the number of detected humans from ith camera.

#### 4.1.2. Appearance Feature Extraction

The edge device then extracts appearance features. To extract appearance features, the edge device collects RGB color samples between the 2D coordinates of the body joints obtained from the 2D human pose detection. For instance, it captures the RGB values between the left and right shoulder joints to represent the upper body’s color. Similarly, it extracts RGB samples for the lower body region. After collecting RGB color samples of a person, the edge device makes RGB histograms to identify the most frequently occurring RGB values representing the clothing colors.

Figure 2 shows a process of appearance feature extraction. The left column displays the raw RGB image and 2D human detection results. In the middle column, we present the RGB histogram, which is a quantified analysis of the color distribution. Finally, the right column shows the extracted appearance feature of upper and lower body regions. These features are represented by the most frequently occurring RGB values. As a result, the data size of the raw image is condensed to 6 integers(8 bit), comprising three RGB values for the upper body and three RGB values for the lower body. Here, we denote the appearance feature as follows:(2)$${\left\{a_{i,\;k}\right\}}_{i\in [1,\;N]}^{k\in [1,\;K_{i}]},\;a_{i,\;k}=\left\{R_{up},\;G_{up},\;B_{up},\;R_{low},\;G_{low},\;B_{low}\right\}$$
where N is also the number of edge devices and $K_{i}$ is the number of detected humans from ith camera.

In this paper, we extracted RGB samples for the upper body, specifically between the left shoulder and right shoulder, left shoulder and left hip, and right shoulder and right hip. Additionally, we collected RGB samples for the lower body, between the left hip and left knee, and right hip and right knee. We extracted a total of 400 color samples from each body, such as in Figure 3.

### 4.2. Central Server

The central server receives the results, $p_{i,\;k}$ and the corresponding $a_{i,\;k}$ from the N edge devices. The central server makes groups of 2D skeletons by geometrical affinities and appearance similarity to distinguish among multiple persons, respectively. Then, it compares these two groups to check calibration errors. If a camera which has the wrong calibration is identified, the central server employs a perspective-n-point (PnP) solving algorithm for calibration self-correction, ensuring accuracy in the 3D pose reconstruction.

#### 4.2.1. Geometrical Affinity

First, the central server measures geometrical affinity to make groups of 2D human poses corresponding to the same physical 3D point in space, like most previous methods [9,25,27]. The central server calculates the spatial relationships between the detected 2D poses and all the hypotheses in 3D which are geometrically aligned with the 2D poses. When a point is observed by two cameras, we can draw epipolar lines, its projection of the point onto each other image plane. By measuring the distance between the 2D pose and all the epipolar lines, we can make a group of 2D poses which has the minimum cost. Figure 4 shows the epipolar lines for two camera views.

Several approaches have been proposed for grouping 2D human pose detections corresponding to the same individual, including the methods described in [9,25] and others. A notable example is the iterative greedy matching algorithm, renowned for its simplicity [27]. In this work, we adapt the iterative greedy matching algorithm as our baseline for geometric affinity-based matching. Our objective is to evaluate the improvements in matching accuracy and robustness when appearance affinity is incorporated into the process. Thus, we note that the formulas and notations described in this section are derived from the iterative greedy matching algorithm presented in [27]. Additionally, our approach can be applied to all geometric approaches that utilize camera calibration data for cross-view matching [27,28,29].

To estimate the 3D human poses from multi-view camera images, we first associate the detections across all views. We denote that $P_{geo}^{m}$ is the set of 2D human poses that are associated to person m by geometrical affinity. When $\left|P_{geo}^{m}\right|$ exceeds 1, we can reconstruct 3D human pose for person m via triangulation.

Here, the geometrical cost $\Phi_{geo}(p_{i,k},\;P_{geo}^{m})$ for assigning a pose $p_{i,\;k}$ to an existing person candidate $P_{geo}^{m}$ derives from the summation of distances between a pose $p_{i,\;k}$ and epipolar lines projected by $P_{geo}^{m}$.
(3)$$\Phi_{geo}\left(p_{i,k},\;P_{geo}^{m}\right)=\frac{1}{\left|V_{k,l}\right|\left|P_{geo}^{m}\right|}\sum_{p_{j,l}\in P_{geo}^{m}\;}\sum_{v\in V_{k,\;l}}\phi (p_{i,k}(v),\;p_{j,l}(v))$$
where $V_{k,l}$ is the set of joins that are both visible for $p_{i,k}$ and $p_{j,l}$. Also, $p_{i,k}(v)$ denotes the 2D coordinates of joint v of the 2D human pose $p_{i,k}$. The distances between a pose $p_{i,\;k}$ and the epipolar lines projected by $p_{j,l}$ can be defined as follows:(4)$$\phi \left(p_{i,k},\;p_{j,l}\right)=\left|p_{j}^{T}F^{i,j}p_{i}\right|+\left|p_{i}^{T}F^{j,i}p_{j}\right|$$
where $F^{i,j}$ is the fundamental matrix from camera i to camera j. Using the cost function $\Phi_{geo}(p_{i,k},\;P_{geo}^{m})$, we can find $p_{i,\;k}$ associated to an existing person candidate $P_{geo}^{m}$ by solving the bi-partite matching problem as follows:(5)$$X^{*}=\begin{array}{c} argmin\\ X \end{array}\sum_{m=1}^{\left|P_{geo}\right|}\sum_{k=1}^{K_{i}}\Phi_{geo}\left(p_{i,k},\;P_{geo}^{m}\right)X_{k,m}$$
where
$$\sum_{k=1}^{K_{i}}X_{k,m}=1,\;\forall m\;\mathrm{and}\;\sum_{m=1}^{\left|P_{geo}\right|}X_{k,m}=1,\;\forall k$$

If $p_{i,\;k}$ is associated to the person candidate $P_{geo}^{m}$ then $X_{k,m}^{*}$ = 1, otherwise, $X_{k,m}^{*}$ = 0. If $p_{i,\;k}$ is considered as the person candidate $P_{geo}^{m}$ and the distance $\Phi_{geo}\left(p_{i,k},\;P_{geo}^{m}\right)$ are under the threshold $\sigma_{geo}$, $p_{i,\;k}$ is added to the person candidate $P_{geo}^{m}$. Algorithm 1 describes the grouping of 2D human poses with the geometrical affinity proposed in [27].
**Algorithm 1:** Grouping 2D poses with geometrical affinity1:**initialize** $P_{geo}=\left\{\left\{p_{1,\;k}\right\}\right\}$2:**for** camera i← 2 **to** N **do**3: **for** pose k← 1 **to** $K_{i}$ **do**4:  **for** hypothesis m← 1 **to** $\left|P_{geo}\right|$ **do**5:   
$C_{k.m}=$
$\Phi_{geo}\left(p_{i,k},\;P_{geo}^{m}\right)$
6:  
**end**
7: 
**end**
8: 
$X^{*}=\begin{array}{c} argmin\\ X \end{array}\sum_{m=1}^{\left|P_{geo}\right|}\sum_{k=1}^{K_{i}}\Phi_{geo}\left(p_{i,k},\;P_{geo}^{m}\right)X_{k,m}$
9: **for** k, m **where** $X_{k,m}^{*}$= 1 **do**10:  **If** $C_{k.m}<\sigma_{geo}$ **then**11:   
$P_{geo}^{m}=P_{geo}^{m}\cup \left\{p_{i,k}\right\}$
12:  
**else**
13:   
$P_{geo}=P_{geo}\cup \left\{p_{i,k}\right\}$
14:  
**end**
15: 
**end**
16:**end**

#### 4.2.2. Appearance Similarity

In this paper, we also measured appearance similarity to make groups of 2D human poses. Unlike methods relying strictly on geometrical data, appearance similarity leverages the consistent visual features of individuals, such as clothing color. The central server also evaluates the appearance features transmitted by the edge devices, apart from grouping 2D human poses based on geometrical affinity, as it is more intuitive to assume that a person wearing the same clothes is the same person across the multiple views.

In a similar way, to calculate geometrical affinity, we can associate the detections with appearance similarity. We denote that $A_{m}$ is the set of clothing colors that are associated to person m. The appearance cost $\Phi_{app}(a_{i,k},\;A_{m})$ for assigning a color $a_{i,\;k}$ to an existing appearance candidate $A_{m}$ derives from the summation of the difference between a clothing color $a_{i,\;k}$ and an existing person candidate $A_{m}$. The difference between the two appearances is calculated by Euclidean distance as follows:(6)$$\Phi_{app}\left(a_{i,k},\;A_{m}\right)=\frac{1}{\left|A_{m}\right|}\sum_{a_{j,l}\in A_{m}\;}\left|a_{k,l}-a_{i,k}\right|$$

Similar to geometrical affinity, we can find $p_{i,\;k}$ associated to an existing person candidate $P_{app}^{m}$ by solving the bi-partite matching problem using the cost function $\Phi_{app}\left(a_{i,k},\;A_{m}\right)$.
(7)$$X^{*}=\begin{array}{c} argmin\\ X \end{array}\sum_{m=1}^{\left|A\right|}\sum_{k=1}^{K_{i}}\Phi_{app}\left(a_{i,k},\;A_{m}\right)X_{k,m}$$
where
$$\sum_{k=1}^{K_{i}}X_{k,m}=1,\;\forall m\;\mathrm{and}\;\sum_{m=1}^{\left|A\right|}X_{k,m}=1,\;\forall k$$

If $a_{i,\;k}$ is considered as the appearance candidate $A_{m}$ then $X_{k,m}^{*}$ = 1, otherwise, $X_{k,m}^{*}$ = 0. If $a_{i,\;k}$ is considered as the appearance candidate $A_{m}$ and the difference $\Phi_{app}\left(a_{i,k},\;A_{m}\right)$ are under the threshold $\sigma_{app}$, $p_{i,\;k}$ is added to the person candidate $P_{app}^{m}$ because the result $a_{i,\;k}$ obtained from appearance feature extraction is matched with 2D human poses $p_{i,k}$. Algorithm 2 summarizes the grouping of 2D human poses with appearance similarity. The result of the algorithm is $P_{app}$, which represents the set of 2D human poses associated with individuals based on appearance similarity.
**Algorithm 2:** Grouping 2D poses with appearance similarity1:**initialize** 
$P_{app}=\left\{\left\{p_{1,\;k}\right\}\right\}$
$\;\mathrm{and}\;A=\left\{\left\{a_{1,\;k}\right\}\right\}$
2:**for** camera i← 2 **to** N **do**3: **for** pose k← 1 **to** $K_{i}$ **do**4:  **for** hypothesis m← 1 **to** $\left|A\right|$ **do**5:   
${C^{\prime }}_{k.m}=$
$\Phi_{app}\left(a_{i,k},\;A_{m}\right)$
6:  
**end**
7: 
**end**
8: 
$X^{*}=\begin{array}{c} argmin\\ X \end{array}\sum_{m=1}^{\left|P_{app}\right|}\sum_{k=1}^{K_{i}}\Phi_{app}\left(a_{i,k},\;A_{m}\right)X_{k,m}$
9: **for** k, m **where** $X_{k,m}^{*}$= 1 **do**10:  **If** ${C^{\prime }}_{k.m}<\sigma_{app}$ **then**11:   
$A_{m}=A_{m}\cup \left\{a_{i,k}\right\}$
12:   
$P_{app}^{m}=P_{app}^{m}\cup \left\{p_{i,k}\right\}$
13:  
**else**
14:   
$A=A\cup \left\{a_{i,k}\right\}$
15:   
$P_{app}=P_{app}\cup \left\{p_{i,k}\right\}$
16:  
**end**
17: 
**end**
18:**end**

#### 4.2.3. Multi-View Triangulation and Calibration Self-Correction

In most research, camera calibration is assumed to be ideal. However, in practice, calibration drift or inaccuracies can occur due to environmental factors or hardware movement. In this paper, we propose a calibration self-correction method using two sets, $P_{geo}$ and $P_{app}$, which represent 2D human poses obtained from geometrical affinity and appearance similarity, respectively. This system achieves robust and accurate 3D pose estimation with an additional 6 byte integers representing clothing colors. Figure 5 shows the overall pipeline of multi-view triangulation and calibration self-correction.

After obtaining the 2D human poses, the server reconstructs the 3D human pose from the set of the 2D human poses $P_{geo}$, leveraging the principles of projective geometry and linear algebra. As described, when $\left|P_{geo}^{m}\right|$ are two and more, we can reconstruct 3D human poses for person m via triangulation. The multi-view triangulation technique, facilitated by the Direct Linear Transform (DLT) method [25], is a widely recognized approach in computer vision. We denote $H_{m}$ as 3D human pose for person m.

To identify incorrectly calibrated cameras, we utilize the differences between two sets for person m, $P_{geo}^{m}-P_{app}^{m}$ is the set difference which isolates 2D human poses that are recognized through geometrical affinity but not confirmed by appearance similarity. In essence, it represents geometric matches without appearance confirmation. Conversely, $P_{app}^{m}-P_{geo}^{m}$ is the set difference which isolates 2D human poses that are recognized through appearance similarity but not aligned with geometrical affinity.

If $n\left(P_{geo}^{m}-P_{app}^{m}\right)=0$ for a specific camera but $n\left(P_{app}^{m}-P_{geo}^{m}\right)>0$, it indicates a potential calibration error for the camera. It means the appearance-based method is functioning as expected. However, if there are skeletons identified by clothing color ($P_{app}^{m}$) that are not identified by epipolar geometry ($P_{geo}^{m}$), this discrepancy indicates a failure in the geometric method, likely due to incorrect camera calibration. The server finds the wrong calibrated camera from a set $P_{app}^{m}-P_{geo}^{m}$. The wrong calibrated camera index $i^{*}$ can be defined as follows:(8)$$i^{*}=\left\{i\;|\;\forall p_{i,k}\;\in \;P_{app}^{m}-P_{geo}^{m}\right\}$$

Given a 3D skeleton $H_{m}$ and a 2D skeleton $p_{i^{*},\;k}$ from the miscalibrated camera, we utilized the SolvePnP or SolvePnPRansac algorithm implemented in [34] to estimate the rotation matrix and translation vector. These perspective-n-point (PnP) solving algorithms estimate the camera rotation and translation from 3D–2D point correspondences. Then, these estimated output $R^{\prime }$ and $t^{\prime }$ are applied to correct the calibration of the camera indexed by $i^{*}$. This feedback mechanism enables updates to incorrectly estimated calibration values in real-time, increasing the accuracy of reconstructed 3D pose.

## 5. Experimental Results

### 5.1. Simulation Environment

In this section, we describe the results of our experiments. First, we applied rotations of θ degrees to both the x-axis and y-axis in the camera coordinate system in order to simulate changes in the camera angle caused by external environmental factors, which occurred after the initial camera calibration. Figure 6 shows the miscalibrated camera environment by rotating the camera with θ degrees to both the x-axis and y-axis. These rotations in the camera coordinate system resulted in the following changes to the image coordinate system. In this environment, we verified that the proposed method effectively corrected calibration errors and enhanced accuracy.
(9)$$R_{\theta }=R_{x}R_{y}=\left[\begin{array}{ccc} 1 & 0 & 0\\ 0 & \cos\theta & -\sin\theta \\ 0 & \sin\theta & \cos\theta \end{array}\right]\left[\begin{array}{ccc} cos\theta & 0 & \sin\theta \\ 0 & 1 & 0\\ -\sin\theta & 0 & \cos\theta \end{array}\right]$$

### 5.2. Quantitative Results (PCP Score)

First, we measured the accuracy using the PCP (Percentage of Correct Parts) score for a quantitative comparison [21]. The evaluation compared the accuracy of the 2D human poses grouped by appearance similarity and geometric affinity. Subsequently, we applied our calibration self-correction method to directly compare the effectiveness. Our calibration correction approach can be generalized and applied to all geometrical methods that rely on camera calibration data for cross-view matching [27,28,29]. In this paper, to benchmark our results against the geometrical method, we reimplemented the iterative greedy matching algorithm [27]. In the absence of calibration errors ($\theta =0^{{}^{\circ}}$), the difference between the PCP score reported in [27] and our implementation was less than 1%.

#### 5.2.1. Campus Dataset

Table 1 shows the PCP scores for the Campus dataset [5], which was acquired using three cameras. We applied a calibration error to one of the three cameras and evaluated the results. As expected, the 3D human pose reconstructed using only the 48-bit appearance similarity had the lowest accuracy due to the limited information and challenges like overlaps or individuals wearing similar clothing.

When there was no calibration error ($\theta =0^{{}^{\circ}}$), the geometric approach also performed well. However, the geometric method experienced a significant drop in the PCP scores of about 12%, with a calibration error $\left(\theta =3^{{}^{\circ}}\right)$. Conversely, with a calibration error $\left(\theta =3^{{}^{\circ}}\right)$, our proposed method improved the PCP scores by approximately 11.5% compared to the geometric method, while the accuracy decreased by about 2.2% compared to the geometric method in an ideal environment without calibration errors ($\theta =0^{{}^{\circ}}$).

When comparing the performance across increasing calibration error levels from $0^{{}^{\circ}}$ to $3^{{}^{\circ}}$, the PCP scores of a geometrical approach did not significantly change because the affected camera was geometrically excluded from the pose estimation process.

Figure 7 illustrates the average PCP scores across different frames. The x-axis represents the frame number, while the y-axis depicts the corresponding PCP scores, providing insights into the temporal dynamics of the pose estimation accuracy throughout the video sequence. The pale blue area represents the PCP score improvement achieved through calibration correction.

#### 5.2.2. Shelf Dataset

In the subsequent experiment with the Shelf dataset [5], which was captured using five cameras, we applied calibration errors to two of these cameras. Table 2 shows the PCP scores for the Shelf dataset. Similar to the findings with the Campus dataset, the geometric method saw a substantial decrease in PCP scores of around 11.5%, with a calibration error ($\theta =3^{{}^{\circ}}$). Our proposed method improved the PCP scores by approximately 8% compared to the geometric method, despite a slight decrease in accuracy of about 1% compared to the geometric method under ideal conditions. The average PCP scores in Shelf dataset across frames are also shown in Figure 8.

### 5.3. Qualitative Results

Apart from the quantitative evaluations, we also conducted qualitative analyses by visualizing the restored 3D poses. To visualize and compare the performance of our proposed approach with the traditional geometric method, we utilized the Shelf dataset. Figure 9 presents the ground truth 3D poses, reconstructed 3D poses using the geometric method, and our proposed approach, respectively, in a scenario where the camera rotation changed by 3 degrees. This visual comparison shows the improvement in the accuracy of reconstructed 3D poses which is achieved by our method.

In the first and second row of Figure 9, highlighted with red circles, the 3D poses reconstructed using the geometric method are inaccurately represented. It shows the limitations of this approach in handling calibration errors. In contrast, the same figure demonstrates that our proposed method successfully recovers the 3D pose.

Furthermore, in the third row of Figure 9, the geometric method incorrectly restored the scene as if eight people were present rather than the four people truly recorded by the camera. In contrast, our proposed approach accurately estimated the correct number of 3D poses to be four, matching the actual video scene. This overestimation of the number of people by the geometric method shows the significant impact of calibration inaccuracies on the reliability of geometric reconstruction methods. Our approach was able to correctly estimate the true number of individuals in the scene by incorporating appearance information and calibration error handling.

## 6. Conclusions

This paper proposes a noise-robust 3D pose estimation framework for real-world scenarios, such as when the data transmission bandwidth is limited and the pose of a mounted camera is vulnerable to external disturbances, including shock, vibration or wind. We consider an edge processing system that consists of multiple edge devices working with a single central server to use essential information instead of raw images with huge sizes. The edge devices transmit the 2D human pose data to the central server, including an additional 48 bits of appearance features. Then, the central server groups the 2D human poses corresponding to each individual and detects any camera with an incorrect calibration to apply calibration error correction. To identify the camera which has the wrong parameter, the proposed framework compares two sets of images, which represent the 2D human poses obtained from geometrical affinity and appearance similarity, respectively. If a discrepancy is present, this could be a signal about the calibration error of a specific camera because it means that the same person based on appearance appears to be a different person geometrically. Then, we estimate the rotation matrix and translation vector via the perspective-n-point (PnP) solving algorithms. The estimated rotation matrix and translation vector are then applied to the wrongly calibrated camera. This feedback mechanism enables updates to the incorrectly estimated calibration values in real-time, increasing the accuracy of reconstructed 3D poses. In addition, we analyzed the experimental results, including PCP score comparisons and visual inspections of restored 3D poses, to verify the improvement in the proposed framework by adopting appearance similarity. We demonstrated that our approach, which combines appearance and geometric data, increased the accuracy of the 3D pose estimation. It achieved around an 11.5% and 8% improvement in the accuracy of 3D pose estimation on the Campus and Shelf datasets, respectively.

## Figures and Tables

**Figure 1 sensors-24-05645-f001:**
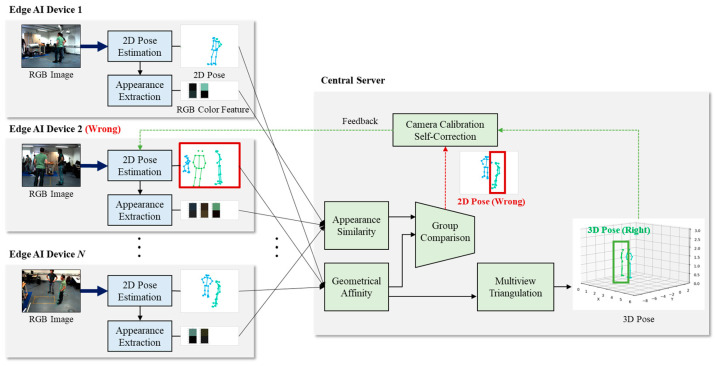
The overall system architecture of proposed framework.

**Figure 2 sensors-24-05645-f002:**
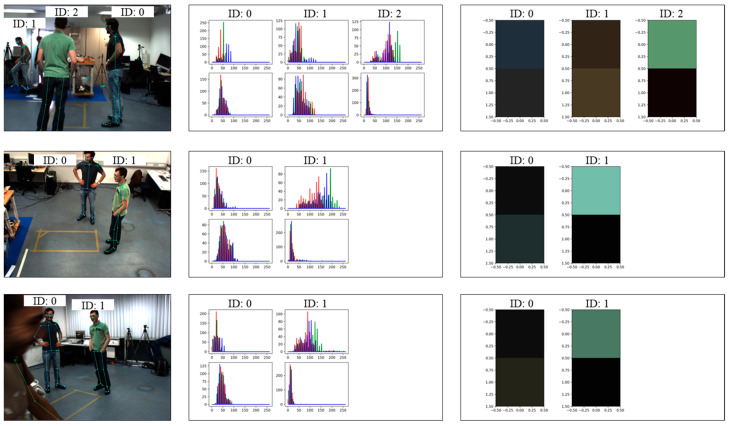
Appearance feature extraction from RGB image.

**Figure 3 sensors-24-05645-f003:**
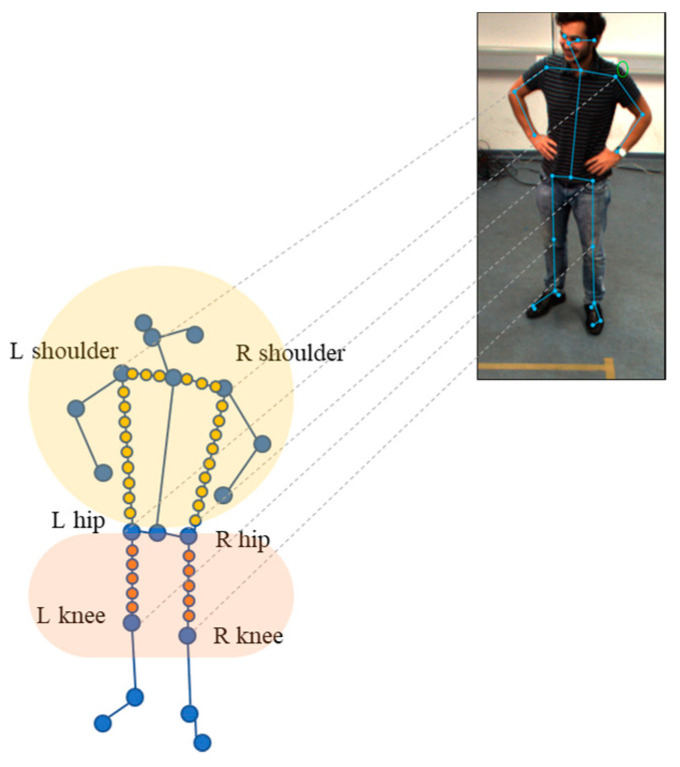
RGB color extraction from 2D skeleton.

**Figure 4 sensors-24-05645-f004:**
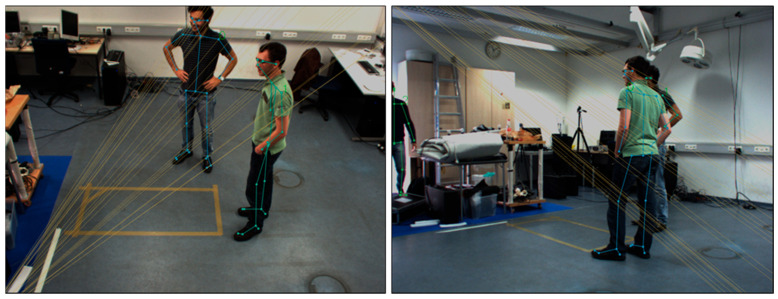
Epipolar lines through for two camera views.

**Figure 5 sensors-24-05645-f005:**
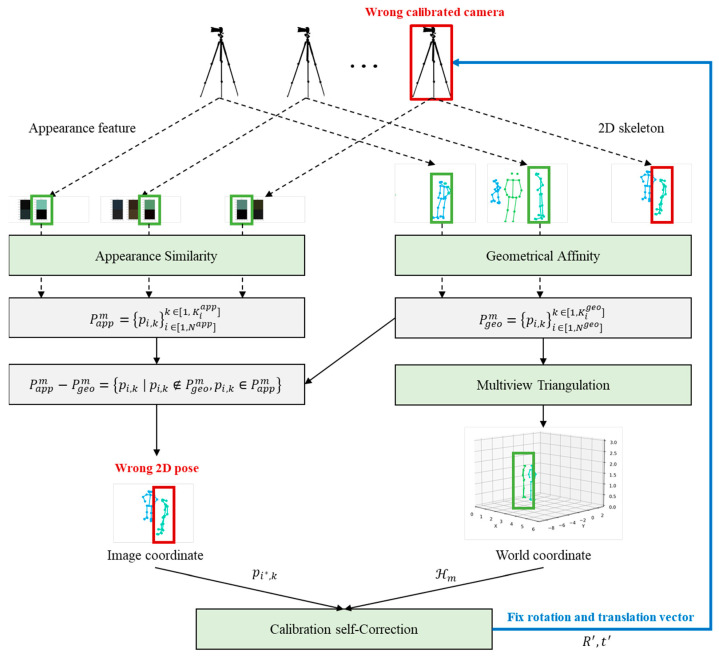
Overall pipeline of multi-view triangulation and calibration self-correction.

**Figure 6 sensors-24-05645-f006:**
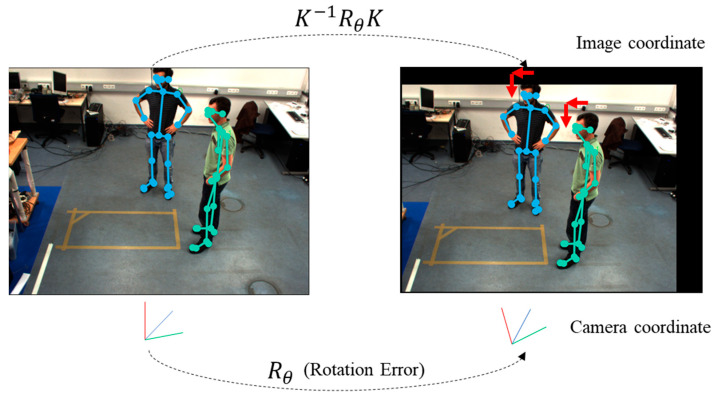
Miscalibrated camera environment by rotating the camera with angle θ.

**Figure 7 sensors-24-05645-f007:**
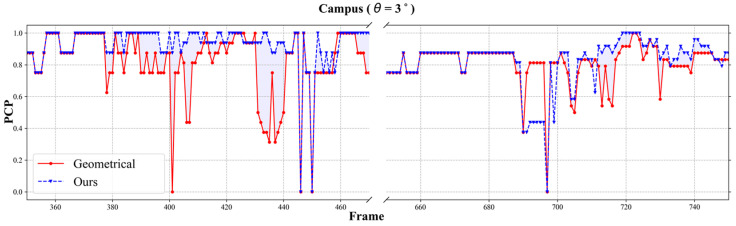
Average PCP score over frame in Campus dataset ($\theta =3^{{}^{\circ}}$); geometrical-method [27] and ours.

**Figure 8 sensors-24-05645-f008:**
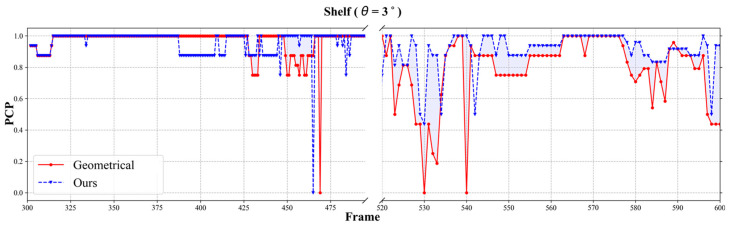
Average PCP score over frame in Shelf dataset ($\theta =3^{{}^{\circ}}$); geometrical-method [27] and ours.

**Figure 9 sensors-24-05645-f009:**
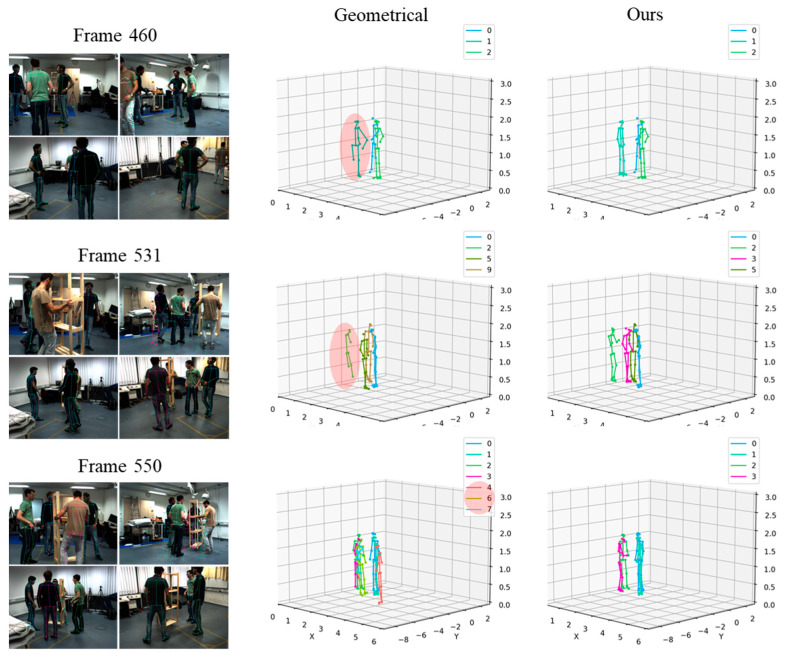
Reconstructed 3D poses with calibration error ($\theta =3^{{}^{\circ}}$); geometrical-method [27] and ours.

**Table 1 sensors-24-05645-t001:** PCP score in Campus dataset.

	$\theta =0^{{}^{\circ}}$				$\theta =1^{{}^{\circ}}$			
Actor	**1**	**2**	**3**	**Avg.**	**1**	**2**	**3**	**Avg.**
Appearance	0.88	0.88	0.45	0.73	-	-	-	-
Geometrical [27]	0.88	0.89	0.89	0.89	0.66	0.81	0.85	0.77
Ours	0.83	0.89	0.89	0.87	0.83	0.9	0.89	0.88
	$\theta =2^{{}^{\circ}}$				$\theta =3^{{}^{\circ}}$			
Actor	**1**	**2**	**3**	**Avg.**	**1**	**2**	**3**	**Avg.**
Appearance	-	-	-	-	-	-	-	-
Geometrical [27]	0.66	0.82	0.85	0.78	0.66	0.82	0.85	0.78
Ours	0.83	0.9	0.89	0.88	0.83	0.9	0.89	0.88

**Table 2 sensors-24-05645-t002:** PCP score in Shelf dataset.

	$\theta =0^{{}^{\circ}}$				$\theta =1^{{}^{\circ}}$			
Actor	**1**	**2**	**3**	**Avg.**	**1**	**2**	**3**	**Avg.**
Appearance	0.80	0.40	0.62	0.61	-	-	-	-
Geometrical [27]	0.98	0.91	0.95	0.95	0.94	0.82	0.95	0.9
Ours	0.97	0.91	0.94	0.94	0.98	0.9	0.96	0.94
	$\theta =2^{{}^{\circ}}$				$\theta =3^{{}^{\circ}}$			
Actor	**1**	**2**	**3**	**Avg.**	**1**	**2**	**3**	**Avg.**
Appearance	-	-	-	-	-	-	-	-
Geometrical [27]	0.91	0.70	0.93	0.85	0.91	0.70	0.93	0.85
Ours	0.97	0.85	0.83	0.88	0.97	0.85	0.83	0.88

## Data Availability

Data are available in a publicly accessible repository. The data presented in this study are openly available in https://campar.in.tum.de/Chair/MultiHumanPose, reference number [5] (accessed on 29 August 2024).

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
