# Peer review of "Noise-Robust 3D Pose Estimation Using Appearance Similarity Based on the Distributed Multiple Views"

_sensors, 2024, doi:10.3390/s24175645_

Round 1
Reviewer 1 Report
Comments and Suggestions for Authors
1. The method proposed in this paper is technically reasonable. It is a common idea to analyze and aggregate features extracted from different ways to improve the final result. Based on this idea, the method proposed in this paper uses several steps. Although the overall combination form is different from other works, the unique advantages of this combination are not well explained in this paper, and the innovation is not strong. It is suggested to refine the innovation points of the method proposed in this paper.
2. What is the basis for the experiment in the paper, which only uses calibration errors of 0 degrees and 3 degrees? It is suggested to increase the selection of calibration errors to illustrate the robustness of the proposed method.
3. There are many methods for 3D Pose Estimation, and the experimental results in the paper are not adequately compared with the results of recent methods. It is recommended to add relevant comparisons.
4. For reproducibility, it is suggested to make the Shelf Dataset available.
Comments on the Quality of English LanguageCan be improved
Author Response
Comment 1-1: The method proposed in this paper is technically reasonable. It is a common idea to analyze and aggregate features extracted from different ways to improve the final result. Based on this idea, the method proposed in this paper uses several steps. Although the overall combination form is different from other works, the unique advantages of this combination are not well explained in this paper, and the innovation is not strong. It is suggested to refine the innovation points of the method proposed in this paper.
Author response: I accepted the comment and supplemented the contents.
Author action: Thank you for your thoughtful feedback. In response to your suggestion to refine the innovation points of our method, I have revised the introduction and conclusion sections to clearly describe the main objective and contribution of our work.
We think the primary objective of this paper is building a noise-robust system by using additional appearance features based on conventional geometrical approaches. In experimental results, we verified that only 48 bits clothing color can enhance 3D pose estimation accuracy. This approach could be useful in a practical distributed system consisting of edge devices, where minimal data is transmitted to the central server due to the limited bandwidth and computational resources.
These revisions are now reflected in the introduction and conclusion section on pages 2 and 14. We hope these updates clarify the contribution of our work.
____________________________________________________________________________________________________________
Comment 1-2: What is the basis for the experiment in the paper, which only uses calibration errors of 0 degrees and 3 degrees? It is suggested to increase the selection of calibration errors to illustrate the robustness of the proposed method.
Author response: I accepted the comment and supplemented contents. Thank you for your valuable feedback.
Author action: I added Table 1 and 2, which shows the PCP scores for different rotation errors (degree from 0 to 3) in Campus and Shelf dataset. As a result, the PCP scores does not significantly change between rotation error because miscalibrated cameras are excluded by geometrical affinity calculation. However, noise (rotation error) does reduce the overall accuracy of 3D pose estimation as expected and we verified that the proposed scheme improves accuracy.
____________________________________________________________________________________________________________
Comment 1-3: There are many methods for 3D Pose Estimation, and the experimental results in the paper are not adequately compared with the results of recent methods. It is recommended to add relevant comparisons.
Author response: I supplemented the contents. Thank you for your valuable feedback.
Author action: As noted in our paper, geometric approaches within cross-view matching, including the method in [27] as well as [28, 29], are affected by camera calibration errors. To compare the impact of such errors with the recent methods, it would require reimplementation of their algorithms at the code level.
For this reason, we selected the method in [27] as our baseline to evaluate the effectiveness of incorporating appearance similarity for calibration self-correction. By focusing on this baseline, we aimed to demonstrate how our approach can improve accuracy in the presence of camera calibration errors.
To address your concern more thoroughly, we have now supplemented our manuscript with additional explanations that clarify the rationale behind our choice of baseline and how our method enhances performance under calibration error conditions. We hope these additions provide the necessary context and justification for our experimental approach.
____________________________________________________________________________________________________________
Comment 1-4: For reproducibility, it is suggested to make the Shelf Dataset available.
Author response: I accepted the comment and revised the fault.
Author action: I added citation about the paper which provides the Campus and Shelf dataset. In addition, I added a Data Availability Statement at the end of the manuscript. The statement clarifies that the data used in this study are available in a publicly accessible repository.

Reviewer 2 Report
Comments and Suggestions for Authors
1. The author presented a multi-person 3D pose estimation method that utilizes multiple edge devices to detect 2D poses from various locations. A central server then detects each person's pose based on appearance similarity and geometrical affinity, and corrects camera calibration errors to achieve accurate 3D pose estimation. The presentation and the results are clear to the readers.
2. Figure 6 shows how calibration errors are corrected. Since the left and right images are the same, the author assumes that the camera has rotated by angle 𝜃 degrees and corrected the skeleton's position. However, the new skeleton position differs from the person in the image, which may cause confusion for the reader.
Author Response
Comment 2-1: The author presented a multi-person 3D pose estimation method that utilizes multiple edge devices to detect 2D poses from various locations. A central server then detects each person's pose based on appearance similarity and geometrical affinity, and corrects camera calibration errors to achieve accurate 3D pose estimation. The presentation and the results are clear to the readers.
Author response: Thank you for your positive feedback regarding the clarity of the presentation and results in our paper. We have made sure to maintain this clarity throughout the manuscript.
____________________________________________________________________________________________________________
Comment 2-2: Figure 6 shows how calibration errors are corrected. Since the left and right images are the same, the author assumes that the camera has rotated by angle ? degrees and corrected the skeleton's position. However, the new skeleton position differs from the person in the image, which may cause confusion for the reader.
Author response: I accepted the comment and supplemented contents. Thank you for your valuable feedback.
Author action: Figure 6 shows a miscalibrated camera scenario within a simulation environment. To enhance clarity, I have added subtitle “5.1. Simulation Environment” and revised both figure 6 and its description. I hope these changes enhance the clarity and eliminate potential confusion.
____________________________________________________________________________________________________________

Reviewer 3 Report
Comments and Suggestions for Authors
This paper proposes an 3D pose estimation using multiple cameras. Overall quality of this paper is quite excellent. However, the reviewer recommend the paper to be revised as follows.
1. Texts in Figures 2, 7, 8 are too small to be read, so it should be magnified.
2. Photos in figure 8 are too small to be read, so it should be magnified.
3. It would be better to show the pose estimation accuracy vs. noise amount, but it is not mandatory.
Author Response
Comment 3-1: Texts in Figures 2, 7, 8 are too small to be read, so it should be magnified.
Author response: I accepted the comment and supplemented contents. Thank you for your valuable feedback.
Author action: I enlarged the text in Figures 2, 7, and 8 for the readability.
These changes can be found on pages 6, 12, and 13 of the revised manuscript.
____________________________________________________________________________________________________________
Comment 3-2: Photos in figure 8 are too small to be read, so it should be magnified.
Author response: I accepted the comment and supplemented contents. Thank you for your valuable feedback.
Author action: The photos in Figure 8 have been magnified to improve visibility and clarity for the readers.
____________________________________________________________________________________________________________
Comment 3-3: It would be better to show the pose estimation accuracy vs. noise amount, but it is not mandatory.
Author response: I accepted the comment and supplemented contents. Thank you for your valuable feedback.
Author action: I added Table 1 and 2, which shows the PCP scores for different rotation errors (degree from 0 to 3) in Campus and Shelf dataset. As a result, the PCP scores does not significantly change between rotation error because miscalibrated cameras are excluded by geometrical affinity calculation. However, noise (rotation error) does reduce the overall accuracy of 3D pose estimation as expected and we verified that the proposed scheme improves accuracy.

Round 2
Reviewer 1 Report
Comments and Suggestions for Authors
No comment
Comments on the Quality of English Language
Can be improved